# Comparing rates of introgression in parasitic feather lice with differing dispersal capabilities

Jorge Doña [1,2✉], Andrew D. Sweet[1,3] & Kevin P. Johnson [1✉]

Organisms vary in their dispersal abilities, and these differences can have important biological consequences, such as impacting the likelihood of hybridization events. However, there is still much to learn about the factors influencing hybridization, and specifically how dispersal ability affects the opportunities for hybridization. Here, using the ecological replicate system of dove wing and body lice (Insecta: Phthiraptera), we show that species with higher dispersal abilities exhibited increased genomic signatures of introgression. Specifically, we found a higher proportion of introgressed genomic reads and more reticulated phylogenetic networks in wing lice, the louse group with higher dispersal abilities. Our results are consistent with the hypothesis that differences in dispersal ability might drive the extent of introgression through hybridization.

[1] Illinois Natural History Survey, Prairie Research Institute, University of Illinois at Urbana-Champaign, 1816 S. Oak St., Champaign, IL 61820, USA.
[2] Departamento de Biología Animal, Universidad de Granada, 18001 Granada, Spain. [3] Department of Entomology, Purdue University, 901 W. State St., West Lafayette, IN 47907, USA. ✉email: jorged@illinois.edu; kpjohnso@illinois.edu

Dispersal is the permanent movement of organisms away from their place of origin. It is a fundamental process in biology with major implications at multiple scales of organization[1–4], including the reproduction of individuals, the composition of populations and communities, and the geographical distribution of species[2,5].

Organisms differ in their dispersal abilities, and these differences have an impact on their biology, such as on the distributional range of a species or gene flow between populations[6]. For example, organisms with lower dispersal abilities tend to have smaller distributional ranges and populations that are genetically more structured[6–8].

Dispersal ability might also affect the opportunities for hybridization between species because the rates at which individuals encounter different species are likely to be higher in organisms with higher dispersal capabilities. Indeed, recent evidence supports this prediction by demonstrating that range expansion is associated with the extent of introgression[9,10]. Similarly, dispersal differences explain more than 30% of the variation in the width of hybrid zones across animals[11]. However, overall there is still much to learn about the factors influencing hybridization[12–14], and, in particular, the influence of dispersal ability on the rate of hybridization remains understudied.

Testing for the effect of dispersal on hybridization should ideally hold constant most factors other than dispersal. The ecological replicate system of wing and body lice (Insecta: Phthiraptera) of pigeons and doves (Aves: Columbidae) has proven to be an ideal system for comparing the impact of dispersal differences on other aspects of biology, such as population structure and codivergence[7,15–18]. Specifically, this is an excellent system in which to assess the effect of differences in dispersal capabilities on levels of introgression because both of these lineages of feather lice: (1) drastically differ in their dispersal ability[19–21], (2) co-occur across the diversity of pigeons and doves, and (3) have the same basic life history and diet[15,18,22]. Both wing and body lice disperse vertically between parents and offspring in the nest. However, wing lice can also attach to and hitchhike on hippoboscid flies to disperse phoretically between host individuals or host species[19–21]. Indeed, this hitch-hiking dispersal mechanism profoundly influences their degree of population structure and cophylogenetic history[7,16,18,23]. In addition, wing lice have a higher rate of host-switching[15,16,23] (i.e., successful colonization of new host species) and of straggling[24] (i.e., dispersal to new host species without reproduction on that new host).

To compare differences in the extent of introgression between wing and body lice, we used whole-genome data from 71 louse individuals belonging to five species of wing lice (*Columbicola*) and seven species of body lice (*Physconelloides*), that occur across the same suite of host species and have highly comparable patterns of diversification[18,22]. Specifically, both lineages within these two groups of lice that are the focus of this study originated on the common ancestor of *Metriopelia* doves (11.3–14.9 mya) and have a correlated pattern of codiversification within the same group of hosts (including a shared cospeciation event which occurred within the *Metriopelia* genus 5.2–7.4 mya[18,22]). We predicted that wing lice, which have higher dispersal abilities and thus higher odds of encountering individuals of a different louse species on the same host, should show more extensive evidence of introgression (Fig. 1).

## Results

Both approaches revealed highly concordant results: higher levels of introgression among species of wing lice compared to body lice. In particular, using a read-mapping based method, the genomic signature of introgression was significantly higher in wing louse species than in body louse species (GLM with the mean values of the simulations; $F = 21.0705$, df = 69, $P = 2.367 \times 10^{-5}$, $R^2 = 0.58$; Fig. 2 and Table S1, Figs. S1–S12 in the Figshare repository[25]). The contigs assembled from reads mapping to the nonfocal species were in the size range of the loci used as reference (mean max contig length = 1214 bp; mean contig length = 292 bp; Table S3 at Figshare[25]). Even though wing lice showed more evidence for introgression, one body louse individual (included in the GLMs) exhibited the highest level of introgression (Fig. 2 and Figs. S1–S12 at Figshare[25]). However, the other individual from the same taxon, inhabiting the same host species and collected in the same geographic region, did not show these elevated levels of introgression (Table S2, Figs. S1–S12 at Figshare[25]).

Secondly, in a phylogenetic network framework, the optimal networks of wing lice were more reticulated than those of body lice even though the number of taxa included in the networks was

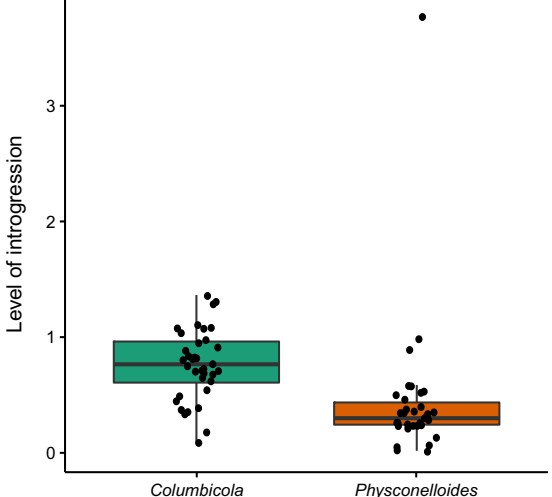

**Fig. 2 Boxplot showing the differences in levels of introgression between wing (green) and body (orange) lice.** Level of introgression represents the sum of the mean coverage of reads mapped from all the species excluding the focal louse species, divided by the mean coverage of the focal louse species (see "Methods" section). Black dots show the levels of introgression (i.e., resulting from the equation) for each individual sample (horizontally jittered values). n = 71 biologically independent louse samples.

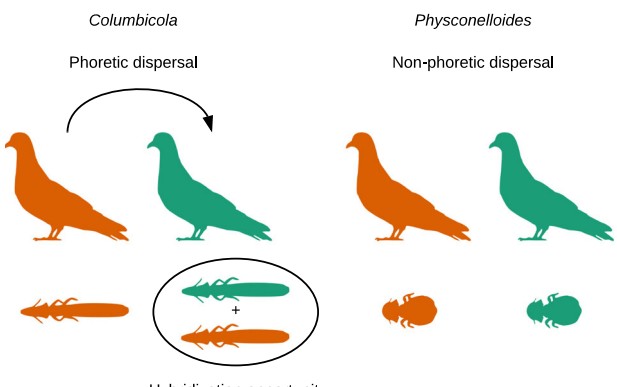

*Columbicola*  —  Phoretic dispersal  |  *Physconelloides*  —  Non-phoretic dispersal

Hybridization opportunity

**Fig. 1 Diagram depicting the ecological replicate system and the hypothesis of this study.** Wing lice (*Columbicola*) have higher dispersal abilities than body lice (*Physconelloides*), and thus higher odds of encountering individuals of a different louse sp.

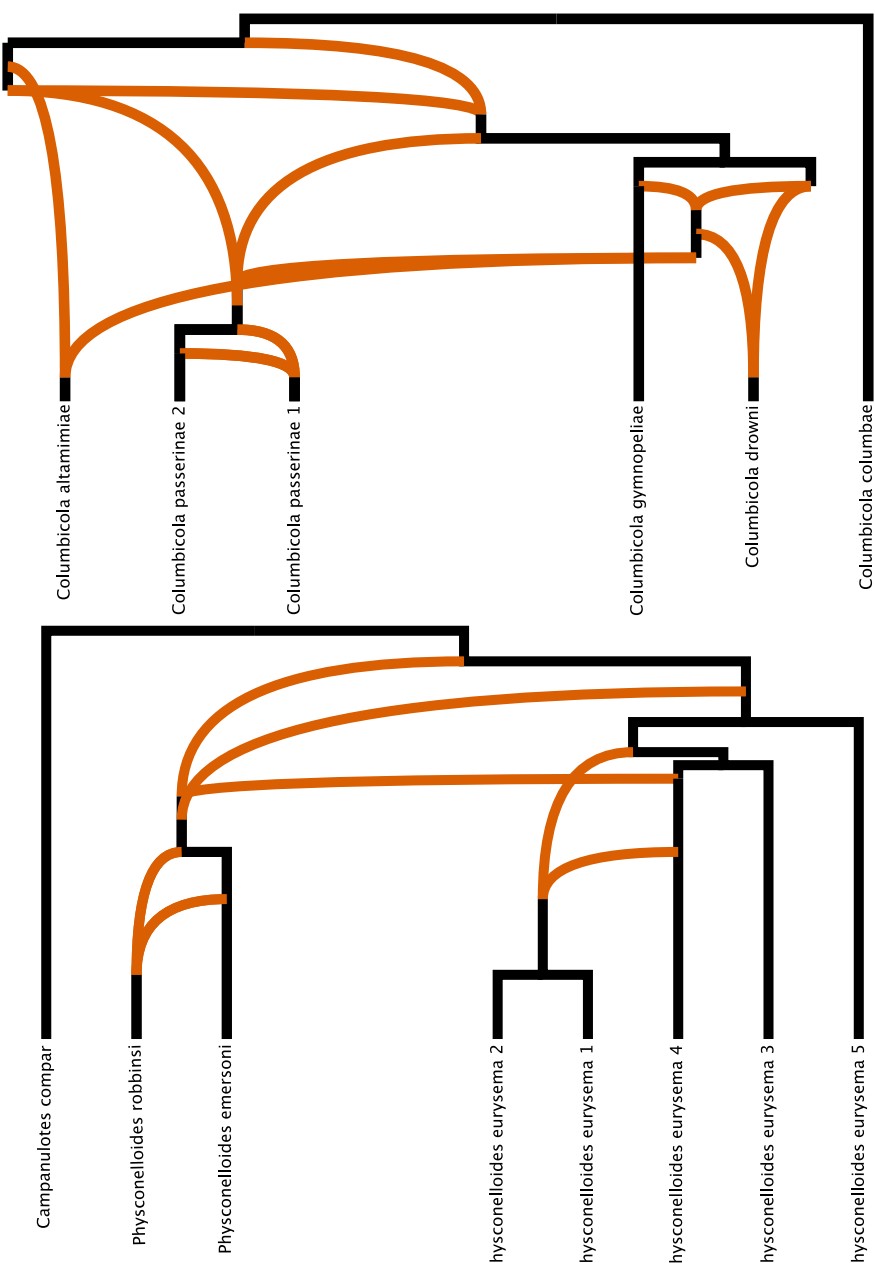

**Fig. 3 Optimal phylogenetic networks of feather lice genera.** Orange branches depict reticulations: seven in *Columbicola* and four in *Physconelloides*.

lower (seven reticulations in *Columbicola* vs. four in *Physconelloides*, Fig. 3). Accordingly, the number of reticulations given the number of potential combinations under a one-tailed test was significantly higher in *Columbicola* (One-sided: $\chi^2 = 3.8132$; df = 1; $P = 0.03$; CI = 0.03–1), and the *P*-value was still near 0.05 with a two-tailed test (Two-sided: $\chi^2 = 3.8132$; df = 1; $P = 0.05$; CI = −0.01–0.66). Also, the specific lineages involved in the reticulations were generally congruent with signatures of introgression from the read-mapping based approach (Figs. S1–S12 at Figshare[25]).

## Discussion
Estimates of introgression in two groups of ectoparasites that differ in their dispersal abilities, wing and body lice of doves, indicate that the lineage with higher dispersal ability (wing lice) shows more evidence of introgression. This evidence from wing and body louse genomes is consistent with the hypothesis that

dispersal differences might drive differences in the level of introgression in this system of parasites. Admittedly, there may be some unknown factor, other than dispersal, differing between these two groups of lice that causes the difference in the level of introgression, but prior work on these groups of parasites points to dispersal as a crucial factor underlying many of the ecological and evolutionary patterns in these parasites. Further research on other taxa is needed to confirm the generality of these findings. This work is among the first studies of introgression in a host-symbiont system[26]. Notably, recent studies have found that straggling and host-switching are relatively common processes in host-symbiont systems[27–30]. Our study suggests that in a straggling/host-switching scenario, hybridization can provide further genetic variation with important ecological and evolutionary consequences (e.g., facilitating adaptation to current hosts or facilitating the colonization of new ones)[31]. Indeed, we may have found a potential recent hybridization event (i.e., the *Physconelloides* individual showing the highest level of introgression),

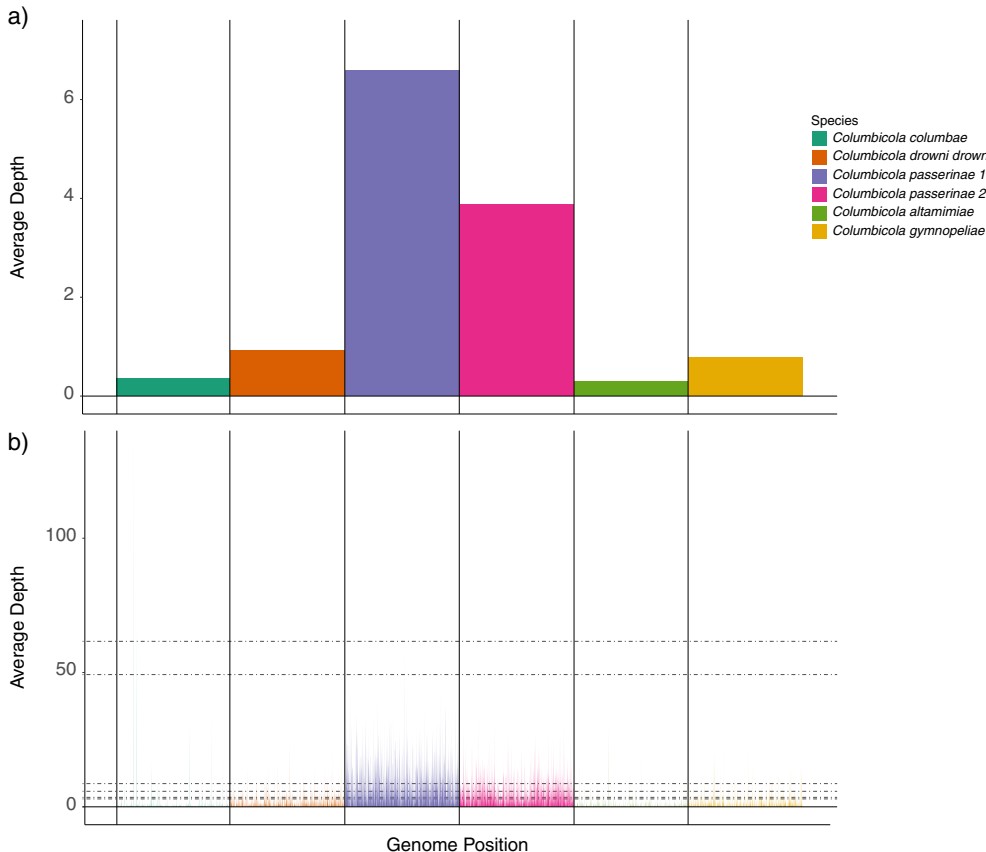

**Fig. 4 Illustrative example of sppIDer results with an individual of Columbicola passerinae 1 (Cosp.Copas.11.7.2016.8). a** The average mean coverages of reads mapping to every species (i.e., the values that were used for the calculations of introgression levels), and **b** shows the mean coverages of reads mapping to each species across the whole set of loci. These and additional visualizations of all the individuals can be found in the Figshare repository[25] (Figs. S1–S12).

though this requires further study to rule out methodological issues (e.g., wet-lab contamination).

In this vein, a careful examination of the introgression history of these taxa (and symbionts as a whole) is needed to better understand the patterns of introgression that we found. Questions such as how much introgression can be expected or how the introgressed regions are retained in parasite/symbiont genomes across time, among many others, require further attention. For instance, in this study, the levels of introgression detected by the sppIDer analyses (i.e., the magnitude but not the comparative pattern) may be unrealistic. It may be that some fraction of the level of introgression detected by sppIDer may be due to ILS, and not introgression. However, both louse genera are expected to have relatively similar rates of ILS (if any). It is also possible that taxon age and interspecific divergence might affect introgression rates. Nevertheless, Sweet and Johnson[18] compared the degree of genetic divergence of two pairs of species of both genera that inhabits the same host species and share a cospeciation event. In this case of taxa of the same age, the pair of *Columbicola* species had lower interspecific genetic distances than *Physconelloides*. This could be as a result of mutation rate differences between the two genera. However, this could also be due to higher gene flow among host infrapopulations due to higher dispersal capabilities of wing lice. The same pattern can be found overall across the species studied here, i.e., on average, lower uncorrected interspecific genetic distances among *Columbicola* than among *Physconelloides* species, though the range of the interspecific distances does overlap (Table S4 at Figshare[25]). Thus, if present, ILS could potentially be more prevalent in *Columbicola* species[32,33]. The

PhyloNet analysis, however, does control for ILS, and showed highly congruent results. In addition, some individual gene trees exhibit signatures suggestive of introgression with highly similar sequences shared by some individuals of different species, and much less likely to be a consequence of ILS (Figs. S13–S14). Overall, the species of *Columbicola* and *Physconelloides* are from the same group of hosts and thus are overall comparatively similar in levels of divergence, so it seems unlikely that these small differences are driving the results.

Another caveat is that sppIDer can detect introgression from species that are not included in the reference data. In those cases, the reads may map to the closest taxon available in the reference set, and thus could artificially increase the level of introgression from a given species[34]. Accordingly, the levels of introgression detected by sppIDer in certain species could be an aggregate of introgression events from more than one species. Indeed, our PhyloNet analysis supports this scenario, with several reticulations from ghost lineages and species (Fig. 3). However, in this system we have nearly complete sampling of host taxa and are missing few, if any, extant species making this concern less likely.

## Methods
**Data**. We analyzed Illumina whole genome sequence data (150 or 160 bp paired-end reads) from 71 louse individuals belonging to five and seven taxa of *Columbicola* and *Physconelloides*, respectively (Table S2 at Figshare[25]) hosted by the monophyletic clade of small New World ground doves. This paper's taxonomic classification of lice is based on Sweet and Johnson[18] species delimitation analyses. In particular, they found most *Columbicola* OTUs matched known species, and some *Physconelloides* OTUs were yet to be formally described as species (and are named here following Sweet and Johnson[18]). All raw sequence data used were

available from previous studies[18,35,36] and represent all described New World ground-dove wing and body louse species (sampled from most host species in this group) including lice samples across multiple biogeographic areas within species of hosts[18] (Table S2 at Figshare[25]). Illumina genome sequence data preprocessing included several steps[18]. First, we discarded duplicate read pairs using the fastqSplitDups script (https://github.com/McIntyre-Lab/mcscriptand https://github.com/McIntyre-Lab/mclib). We then eliminated the Illumina sequencing adapters with Fastx_clipper v0.014 from the FASTX-Toolkit (http://hannonlab.cshl.edu/fastx_toolkit). Also, we removed the first 5 nt from the 5′ ends of reads using Fastx_trimmer v0.014 and trimmed bases from the 3′ ends of reads until reaching a base with a phred score ≥28 (which is equivalent to a base call accuracy higher than 99.8%) using Fastq_quality_trimmer v0.014. Finally, we removed any reads less than 75 nt and analyzed the cleaned libraries with Fastqc v0.11.5 to check for additional errors. We assembled nuclear loci in aTRAM following previous studies[18,36,37]. In particular, we mapped modest coverage (25–60×), multiplexed genomic data to reference loci from a closely related taxon. For our reference set of nuclear loci for wing lice, we used 1039 exons of Columbicola drowni[36] (raw data: SRR3161922). This data set was assembled de novo[37] using orthologous protein-coding genes from the human body louse genome (Pediculus humanus humanus[38] as a set of target sequences. We mapped Columbicola reads to the C. drowni references using Bowtie2[39]. For body lice, we obtained nuclear data using the same pipeline and software parameters, except that we used 1095 loci from Physconelloides emersoni as the reference for mapping. To generate the input ultrametric gene trees for Phylonet v3.6.8[40–42], we first aligned each nuclear locus in MAFFT[43] (--auto) and removed columns with only ambiguous sequences ("N"). Then, we estimated gene trees in RAxML v8.1.3[44] with a GTR + Γ substitution model for each gene alignment. Finally, we made trees ultrametric using the nnls method in the force.ultrametric function within the "phytools" R package[45].

**Quantifying introgression**. We used two different approaches to quantify differences in the extent of introgression (i.e., ancient plus recent) between the two louse genera. We employed methods suitable to detect introgression between species and between individuals from the same species (i.e., we did not employ methods aimed to detect differences at the population level, e.g., TreeMix[46]). First, we used sppIDer[34] to quantify the genomic contributions of different louse species in an individual louse genome (Fig. 4). We built our reference for each genus using all the nuclear loci from a single individual per species. For the reference, we selected those individuals for which we assembled the highest number of genes for each genus. We estimated the extent of introgression as the sum of the mean coverages of reads mapped from all the species excluding the focal louse species, divided by the mean coverage of the focal louse species (Fig. 4). Note that these mean coverage values are calculated using only those reads that mapped with a mapping quality (MQ) > 3[34,47] (Figs. S1–S12). Plots of coverage across the genomes suggested that reads mapping to other species were not artificial mappings (e.g., high coverage mappings to short repetitive regions; Figs. S1–S12). In addition, we used SPAdes v3.12.0 (default parameters) to perform a de novo assembly of the putatively introgressed reads detected by sppIDer.

Second, we quantified introgression at the species level, while accounting for incomplete lineage sorting (ILS) using a maximum pseudo-likelihood framework with PhyloNet 3.6.1[40–42]. Reticulations in this method can be attributed to hybridization events. We trimmed the unrooted gene trees to the same individuals used as reference taxa in sppIDer, and performed eleven independent analyses with a differing maximum number of reticulation nodes (i.e., from zero to ten). We conducted ten runs per analysis. We then selected the optimal network for each genus based on AIC values and slope heuristics.

**Statistics and reproducibility**. We compared the sppIDer results using generalized linear models (GLMs). We used a Gaussian distribution of errors and an identity link function. We performed one GLM for each simulation iteration using the glm function of the "stats" R package[48]. The extent of introgression for each louse genus was the dependent variable, the genus identity was the independent variable, and we accounted for the introgression differences between louse species including louse identity as a fixed factor. We confirmed assumptions underlying GLMs by testing the normality of regression residuals for normality against a Q–Q plot. We also considered the possibility that some of the reads mapping to other species were technical contaminations, i.e., due to index-swapping[49–52]. Previous studies have found that the misassignment of reads generally ranges from 1 to 9%[49–52]. Thus, to account for possible contaminants, we wrote a simulation in R that randomly subtracted 9%[49–52] from the mean coverage value of a particular sample (i.e., we subtracted a random proportion of the mean coverage value for each sample until reaching 9%). We ran 100 iterations of the simulation and ran a GLM for each iteration (Table S1 at Figshare[25]). Finally, we used the $\chi^2$ test to compare the number of species in pairwise comparisons of each genus with the number of reticulations found in each optimal phylogenetic network. Because we had an a priori prediction that Physconelloides should exhibit less evidence of reticulation than Columbicola, we used a one-tailed test; however, we also report the results of the two-tailed test equivalent.

**Reporting summary**. Further information on research design is available in the Nature Research Reporting Summary linked to this article.

## Data availability
All data needed to evaluate the conclusions in the paper are present in the paper and/or the Supplementary Materials published with the paper or in external repositories. Source data (Tables S1–S4, Figs. S1–S14) are available at GitHub (https://jorge-dona.github.io/Comparing-rates-of-introgression-in-parasitic-feather-lice-with-differing-dispersal-capabilities/supplementary.html) and at Figshare (https://doi.org/10.6084/m9.figshare.9176204)[25]. Individual gene trees are available at Figshare[25]. Additional data related to this paper may be requested from the authors.

## Code availability
The code used to account for index-swapping incidence is available at Figshare (https://doi.org/10.6084/m9.figshare.9176204).

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

## Acknowledgements

This study was supported by the US National Science Foundation (DEB-1239788, DEB-1342604, DEB-1926919, and DEB-1925487 to K.P.J.) and the European Commision (H2020-MSCA-IF-2019, INTROSYM: 886532).

## Author contributions

J.D. and K.P.J. conceived the study. J.D., A.D.S., and K.P.J. designed the study. A.D.S. collected the data. J.D. and A.D.S. analysed the data. K.P.J. obtained financial support for the project. J.D. wrote the manuscript and all authors contributed to editing the manuscript.

## Competing interests

The authors declare no competing interests.
