## [Peer Review File · Communications Biology]

Reviewers' comments:

Reviewer #1 (Remarks to the Author):

The authors compare the extent of introgression in the genomes of 35 bird body lice (drawn from 7 taxa) and 36 bird wing lice (drawn from 5 taxa). Wing lice have higher dispersal abilities. They want to test their hypothesis that higher dispersal enhances the chance of introgression.

The authors (1) claim to have evidence that the extent of introgression is significantly higher in wing lice, and (2) go on to claim that this represent a significant step towards supporting their hypothesis. I have difficulties with both of these claims.

(1) The presentation of the evidence for levels of introgression is far from convincing. We are given no information about the degree of divergence among the taxa being compared in each genus. We are given no feeling for the degree of certainty with which "sppIDer" can map reads to different species (the results shown in the supplementary figures). It would be useful to know more about the lengths of the putative introgressed regions. It would be interesting to see some phylogenies for the putative introgressed regions. (And it would be useful to know what the scale is in Figure 2; is this percent?).

The meaning of Figure 3 is quite opaque. For example, on the left, the branch leading to *Columbicola passerinae* 1 is in orange, and so is apparently a "reticulation", yet it connects to a common ancestor with *Columbicola passerinae* 2 (which we might expect to be its closest relative), and there is no other (nonreticulating) branch from *Columbicola passerinae* 1.

(2) The authors suggest that their results "represent an important step for understanding the factors driving hybridization" (page 2), that "differences in dispersal drive differences in the extent of introgression" (page 10), and that their results "represent a significant step towards understanding the factors driving hybridization" (page 11). These claims of the impact of this work are extremely hard to justify.

Even if the data regarding the extent of introgression are accurate, the current data set alone cannot provide a meaningful test of the authors' hypothesis. Since the 7 taxa of body lice all belong to one genus, and the 5 taxa of wing lice all belong to one (other) genus, it is clear that they have only a single phylogenetically independent contrast. It is impossible to conclude a significant association between dispersal and introgression, based on what is (in effect) a single data point. For example, if in reality there were no association between dispersal and introgression, nevertheless given there is an observed difference between body and wing lice, there is a 50% random chance that the difference would be in the right direction to be consistent with the hypothesis.

Reviewer #3 (Remarks to the Author):

The major claim of this paper is that feather lice with higher dispersal abilities hybridize more than lice with lower dispersal abilities, and the implication is that this conclusion likely applies to a broad array of taxa, not just lice. The authors support their claim with two main analyses of whole genome data from several taxa of lice from a high-dispersal genus and a low-dispersal genus: 1) estimating the level of introgression within individual louse genomes and 2) estimating introgression among species. Both analyses show strong support for their conclusion, with

significantly more introgression seen in the high-dispersal louse genus.

I find the results to be both novel and convincing. I'm not aware of previous work using these particular methods to address the question of dispersal ability's effect on hybridization (or really any work quantifying the relationship on a broad-scale genetic level). These two louse genera provide an excellent system for testing the question, and I find it difficult to come up with any reason their results could be explained by some differences in the lice taxa other than dispersal ability. As far as these results being applied to a wider field, I think they absolutely can be. The hypothesized link between dispersal ability and increased hybridization is not limited to lice or their close relatives in any way; it's something that should theoretically apply to any sexual reproducing organisms. My impression is that this paper should become one that anyone studying dispersal ability's effects on speciation/hybridization/natural history should cite, because it provides a novel/rare example of direct quantification of the link between dispersal ability and genetic introgression among species. I think this link is kind of a no-brainer that people don't really doubt must be true in most cases, but now we have a paper that actually provides empirical evidence supporting it. Cool stuff!

Having said that, I think there are some aspects of the paper that should be improved before publication. Foremost is my comment #15, which highlights an anomaly with the chi-squared results. Specific comments, in order by line number:

1) Line 53: "factors influencing hybridization events are poorly known" -- Subjective and debatable. We know quite a bit about hybridization, hybrid zones, reproductive barriers, etc. Something like "we still have much to learn about the factors influencing hybridization..." would likely work better here and not cause significant disagreement from readers that study other aspects of hybridization.

2) Lines 63–69: This is a pretty difficult sentence to follow, particularly because item #2 has a clarifying clause within it. Also, it seems like #3 should have an "and" in front of it if #4 is going to have the "but". Perhaps start with the part about them having different dispersal abilities, then list the similarities? #2 should lose the semicolon and all be one clause as well, maybe eliminating the "cophylogenetic analyses and bird time-calibrated trees indicate that" for brevity and clarity.

3) Lines 81–82: Is it supposed to say "and having highly comparable..."? That, or "that occur across the same host species and have..." The grammar doesn't match up as is.

4) Lines 93–94: "Data were available...": Which data? All the data you used? I honestly was a little confused for a while if you were using some original data along with the data from previous studies. Maybe start the sentence "All sequence data used were..." to be clearer. Additionally, it would be very helpful to briefly describe how the data were obtained in the previous studies instead of just citing them.

5) Lines 95–96: I got confused by what I think is incorrect grammar here. The commas seem to indicate a list, but are you only mentioning louse species and host species (2 items)? Which taxa does the "including samples across..." part refer to? Just lice, just hosts, or both? It needs to be clarified.

6) Line 103: Why 28 as the phred score limit? I confess ignorance of this specific method when asking that question, but I feel like I wouldn't be the only one wondering. I'm not sure it's really necessary to explain, but it was definitely a question I had while reading.

7) Lines 112–113: Okay, so this part really confused me, because I thought all your data came from previous studies. Which reads were newly generated and which were obtained from GenBank? I thought all were obtained from GenBank. The source(s) of all the data definitely needs to be clarified.

8) Line 115: What does the "P" stand for in "P. emersoni"? Although I'm pretty sure you must mean *Physconelloides*, you've mentioned *Physconelloides* and *Pediculus* prior to this, so it would be good to clarify.

9) Lines 124–125: Could be made clearer by deleting "both" and then inserting "between" in front of "individuals"

10) Line 130: Eliminate the use of "Finally" here because it appears between the "First" and "Second" methods described.

11) Line 133: Spell out ILS because it's the first usage in the paper.

12) Lines 135–136: Not that I think it would necessarily change results at all, but why trim the gene trees at this step instead of just construct the original gene trees with only the references? If you constructed the gene trees with only the reference taxa, would the PhyloNet results be the same?

13) Lines 160–165: This is all redundant from the Methods section and feels like it was copy-and-pasted from a previous version of the Methods section (particularly because ILS is spelled out). It should be deleted from the Results section, in my opinion.

14) Line 175 (Figure 2, and Figs S1–S12): The supplementary figures were really good and helped me understand how the level of introgression was being measured for Figure 2. It might be worth putting one of the *sspIDer* graphs into the main paper for that purpose. In regards to the specific results here, some of them seem kind of unreal to me. Several of the *Columbicola* individuals had scores of >1 , which means that over half of the reads from those individuals mapped to different species (if I'm understanding it correctly)? Is that realistic? While the trend between the two genera clearly shows more introgression in *Columbicola* and makes sense, the specific values for introgression from the *sspIDer* results seem odd to me. Perhaps I do not understand the values well enough, but I think they could use some more explanation in the Discussion section. I can't be the only one that would see those results and wonder the same thing about how so much of a genome would map to different species.

15) Lines 183–184: This is the most important result that needs to be addressed before publishing, in my opinion. A chi-squared value of 3.8132 is not significant at the 0.05 level and would have a p-value of 0.05085. Either the p-value or chi-squared value listed here must be incorrect and needs to be fixed. If the chi-squared value is correct, then the discussion needs to be altered accordingly because the result is not as significant (but would still be good support for the hypothesis).

16) Lines 187–188: The third sentence of the caption is awkward. Perhaps eliminate the third sentence and change the second sentence to be "Orange branches depict reticulations, seven in *Columbicola* and four in *Physconelloides*".

17) Lines 196–197: "can provide further variation" -- variation in what?
"important eco-evolutionary consequences" -- Such as?

This sentence could definitely use some fleshing out to be clearer about what you mean.

Overall, I found the paper to be clearly written, aside from some minor sentence structure and grammar issues addressed above. And except for the redundant methods information at the beginning of the Results section, the manuscript does a very good job of being concise. The conclusions are strong without being oversold.

Methods-wise, I think enough detail has been provided to allow for reproduction, and I think the

statistical analyses were sound (aside from the need to address the chi-squared results as reported).

Reviewer #1 (Remarks to the Author):

The authors compare the extent of introgression in the genomes of 35 bird body lice (drawn from 7 taxa) and 36 bird wing lice (drawn from 5 taxa). Wing lice have higher dispersal abilities. They want to test their hypothesis that higher dispersal enhances the chance of introgression.

The authors (1) claim to have evidence that the extent of introgression is significantly higher in wing lice, and (2) go on to claim that this represent a significant step towards supporting their hypothesis. I have difficulties with both of these claims.

>>> We appreciate the critical review of the paper, which has contributed to an improved version of this manuscript. We have endeavoured to modify and improve the manuscript following these comments. Specifically, we have adjusted the scope of the conclusions that can be drawn from our results. Finally, we have also provided additional data to support the evidence of introgression from one of the methods we used (spplDer).

(1) The presentation of the evidence for levels of introgression is far from convincing.

We are given no information about the degree of divergence among the taxa being compared in each genus.

>>> We previously provided information in the Introduction section on how comparable these taxa are in terms of age and divergence times: "both lineages originated on the common ancestor of *Metriopelia* doves (11.3-14.9 mya) and also share a cospeciation event which occurred within the *Metriopelia* genus (5.2-7.4 mya; Sweet and Johnson 2015, 2018)". These louse taxa have been extensively studied in a cophylogenetic and coevolutionary framework (Clayton, Johnson, & Bush, 2016). Based on these studies, we know both groups of lice originated on the same host lineage, and have codiversified in a correlated host-dependent way within the same group of hosts. In particular, they have correlated measures of congruence for individual host-parasite associations (PACO results: no differences, Mann-Whitney U = 57, P = 0.847; but positively correlated, $\rho = 0.71$, P = 0.019; Sweet & Johnson, 2018). Altogether, we are generally confident that these taxa have well-known comparable relative ages. Indeed, we think this scenario of correlated co-diversification makes this system a remarkable natural comparative system in which to test our hypothesis. We have expanded the information on the comparability of these taxa (L65-69).

We are given no feeling for the degree of certainty with which "spplDer" can map reads to different species (the results shown in the supplementary figures). It would be useful to know more about the lengths of the putative introgressed regions.

>>> We understand the reviewer's concerns. First, SpplDer discards reads that mapped with low quality (MQ >3) before calculating the coverage statistics (i.e., the values we used for the introgression analyses), but we did not explicitly state this step in the manuscript. Second, we had the mapping data and different visualizations (e.g., plots of reads mapping to each species across loci) supporting that this signature was driven by a real process and not by something artefactual (e.g., reads mapping with low quality to certain short regions), but we unfortunately had not included this useful information in our original manuscript (and our original supplementary figures did not help clarify things in this regard either because they only show mean coverage values that could have been affected by these artifacts).

In the revised version of our manuscript, we have now provided additional data to support the evidence of introgression from one of the methods we used (sppIDer). Specifically, for each sample, we now have 17 supplementary figures (instead of 1) with different visualizations on the mapping quality of the introgressed regions. We have included: 1) 3 plots on the mapping quality of the reads to each of the reference species, and 2) 13 plots showing introgression signatures across the whole set of loci.

To facilitate the visualization of these 1,207 new plots, we have modified the supplementary Rmarkdown file (Html) that now includes three drop-down pdf files for each individual louse (one encompassing plots with summary values on species depth, one showing coverage across genome plots, and one of mapping qualities). Finally, we have also added to the manuscript information on how sppIDer discard reads mapping with low quality in its further calculations (L 134-136).

We hope these additional data and presentations will clear up concerns regarding potentially artefactual results in our sppIDer analyses.

It would be interesting to see some phylogenies for the putative introgressed regions.

>>> We partially agree here with the reviewer. We do share the interest in the specific introgressed regions; indeed, this question is among the goals of an upcoming Marie Curie project "Conservation impacts of hybridization and introgression in symbionts: Measuring the magnitude and role in shaping eco-evolutionary variables" (INTROSYM; <https://jdona.com/project/introsym/>); however, we believe this point is beyond the scope of this paper, given its complexities. In this study, we were interested in comparing overall rates of introgression and their relationship with differing dispersal capabilities, and not in looking at the particular loci that have introgressed.

On the other hand, these analyses may have been partially done already, though we have not looked specifically at the phylogenies of introgressed regions (for the reasons stated above). Nevertheless, PhyloNet uses the conflict between gene trees topologies (i.e., those from introgressed loci vs. non-introgressed) to infer reticulations, and individual gene trees containing introgressed regions were also part of the analysis.

(And it would be useful to know what the scale is in Figure 2; is this percent?).

>>> It is a numerical scale derived from the ratio of the coverages. Specifically, we calculated the introgression level as the focal species mean coverage relative to the sum of all species (other than the focal) mean coverages. We have now clarified this point in the figure caption.

The meaning of Figure 3 is quite opaque. For example, on the left, the branch leading to *Columbicola passerinae* 1 is in orange, and so is apparently a "reticulation", yet it connects to a common ancestor with *Columbicola passerinae* 2 (which we might expect to be its closest relative), and there is no other (nonreticulating) branch from *Columbicola passerinae* 1.

>>> In this figure, every reticulation involves at least two taxa (extant or extinct). The confusion with the example mentioned (and others) is because the two lines are overlapping. For instance, the orange line towards *Columbicola passerinae* 1 is, in reality, a reticulation between the ancestor of *Columbicola passerinae* 1 and *Columbicola passerinae* 2 (see the raw network of *Columbicola* that we have now included in the supplementary material; Fig S13). This figure is aimed at illustrating the higher number of reticulations of *Columbicola* compared to *Physconelloides* (i.e., complementing figure 1) and was graphically edited to facilitate the visualization by diminishing the noise in the interpretation of this pattern from the raw phylogenetic networks (e.g., showing hybrid speciation events, ghost species, etc.). In the process, we lost clarity for specific links and information, though,

in our opinion, this information is beyond the goals of this manuscript. Nevertheless, as having this information can facilitate the interpretation of reticulations from Figure 3, we have included the raw networks of each taxon as supplementary material, clarified the figure caption text, and explicitly direct readers to Supplementary material for specific details of particular reticulations (L 196-197).

(2) The authors suggest that their results “represent an important step for understanding the factors driving hybridization” (page 2), that “differences in dispersal drive differences in the extent of introgression” (page 10), and that their results “represent a significant step towards understanding the factors driving hybridization” (page 11). These claims of the impact of this work are extremely hard to justify. Even if the data regarding the extent of introgression are accurate, the current data set alone cannot provide a meaningful test of the authors’ hypothesis. Since the 7 taxa of body lice all belong to one genus, and the 5 taxa of wing lice all belong to one (other) genus, it is clear that they have only a single phylogenetically independent contrast. It is impossible to conclude a significant association between dispersal and introgression, based on what is (in effect) a single data point. For example, if in reality there were no association between dispersal and introgression, nevertheless given there is an observed difference between body and wing lice, there is a 50% random chance that the difference would be in the right direction to be consistent with the hypothesis.

>>> We recognize that in the previous version, we may have overstated some of our conclusions (e.g., “differences in dispersal drive differences in the extent of introgression”). Accordingly, we have revised the text to address this issue by avoiding overstating the conclusions. We also agree that our hypothesis would benefit from additional analyses in unrelated species/groups (and we now explicitly acknowledge that in the ms; L 207-207). On the other hand, we disagree with that 1) our study does not represent an important step for understanding the factors driving hybridization, nor that 2) our data set cannot provide a meaningful test of our hypothesis for the following reasons:

1) In this study, if the results would have gone in the opposite direction (or no relationship), we could have ruled out the hypothesis that dispersal differences are important in this system for hybridization (and thus potentially not important in other systems). Consequently, we think we can state that our results are “consistent” with the hypothesis that dispersal differences might drive differences in introgression levels, even though they do not “prove” it.

2) To our knowledge, this is the first time that this hypothesis (which certainly may have an important impact on different areas of research, such as coevolutionary biology) has been tested. Indeed, reviewer 2 also places value in this aspect of the study, e.g., “My impression is that this paper should become one that anyone studying dispersal ability’s effects on speciation/hybridization/natural history should cite.”

3) This hypothesis and its predictions are not theoretically unsupported but arise from extensive knowledge available on these two groups of lice (e.g., ecology, coevolution, etc. Clayton, Johnson, & Bush, 2016). This background knowledge is one of the main reasons we selected this system for investigation. In other words, we went into this study with a strong *a priori* hypothesis based on the extensive background in ecology and evolution in this system. Indeed, the results of our analyses are consistent with what has been found for straggling and host-switching rates, population genetic structure, cophylogenetic history, inbreeding rates, etc., and as also reviewer 2 states, given this current knowledge, it would be harder to explain these differences in introgression levels by any other variable than by dispersal abilities.

4) In some of the statistical analyses of our study, every species has contributed to the relationship between dispersal and the level of introgression. In other words, we do not simply have a single data point with an average for each genus, but every species (for which we have several

individuals/replicates) matters when investigating the relationship between dispersal and introgression level. Specifically, in the GLM based approach, while our predictor was a binary variable (i.e., genus) we included the term species as a fixed factor, i.e., we accounted for the variance in differences of introgression levels that was explained by species (for which we have several individuals/replicates so that to have a reasonable estimate). In other words, our results can be understood as "after accounting by differences in introgression level in each genus explained by the species identity, we found significant differences between the two genera." Accordingly, in this approach, every species has had the opportunity to have a particular introgression level and contribute with it to the result of our analysis. Admittedly the differences in introgression we observed may be due to some other unknown underlying factor that differs between the two groups of lice, and we now acknowledge this in the manuscript. However, we feel that all the available biological evidence points to differences in dispersal as being a key factor driving these differences.

5) Many foundational studies in coevolutionary biology have used a similar approach using these two groups of lice (which comprise a large portion of one of the currently most influential books in the area; "Coevolution of life on hosts integrating ecology and history," Clayton, Johnson, & Bush, 2016). The conclusions derived from these studies, such as that dispersal is an essential driver of many of the ecological and evolutionary patterns, have been later validated in other systems, so we feel that the validity of the study system to address scientific questions has been previously reinforced.

Reviewer #3 (Remarks to the Author):

The major claim of this paper is that feather lice with higher dispersal abilities hybridize more than lice with lower dispersal abilities, and the implication is that this conclusion likely applies to a broad array of taxa, not just lice. The authors support their claim with two main analyses of whole genome data from several taxa of lice from a high-dispersal genus and a low-dispersal genus: 1) estimating the level of introgression within individual louse genomes and 2) estimating introgression among species. Both analyses show strong support for their conclusion, with significantly more introgression seen in the high-dispersal louse genus.

I find the results to be both novel and convincing. I'm not aware of previous work using these particular methods to address the question of dispersal ability's effect on hybridization (or really any work quantifying the relationship on a broad-scale genetic level). These two louse genera provide an excellent system for testing the question, and I find it difficult to come up with any reason their results could be explained by some differences in the lice taxa other than dispersal ability. As far as these results being applied to a wider field, I think they absolutely can be. The hypothesized link between dispersal ability and increased hybridization is not limited to lice or their close relatives in any way; it's something that should theoretically apply to any sexual reproducing organisms. My impression is that this paper should become one that anyone studying dispersal ability's effects on speciation/hybridization/natural history should cite, because it provides a

novel/rare example of direct quantification of the link between dispersal ability and genetic introgression among species. I think this link is kind of a no-brainer that people don't really doubt must be true in most cases, but now we have a paper that actually provides empirical evidence supporting it. Cool stuff!

>>> We thank the reviewer for the careful and highly constructive review, and especially for the positive comments on the originality and quality of this work.

Having said that, I think there are some aspects of the paper that should be improved before publication. Foremost is my comment #15, which highlights an anomaly with the chi-squared results. Specific comments, in order by line number:

1) Line 53: "factors influencing hybridization events are poorly known" -- Subjective and debatable. We know quite a bit about hybridization, hybrid zones, reproductive barriers, etc. Something like "we still have much to learn about the factors influencing hybridization..." would likely work better here and not cause significant disagreement from readers that study other aspects of hybridization.

>>> Agree. Changed (L 25-26; 52-53).

2) Lines 63–69: This is a pretty difficult sentence to follow, particularly because item #2 has a clarifying clause within it. Also, it seems like #3 should have an "and" in front of it if #4 is going to have the "but". Perhaps start with the part about them having different dispersal abilities, then list the similarities? #2 should lose the semicolon and all be one clause as well, maybe eliminating the "cophylogenetic analyses and bird time-calibrated trees indicate that" for brevity and clarity.

>>> Done (L 63-70).

3) Lines 81–82: Is it supposed to say "and having highly comparable..."? That, or "that occur across the same host species and have..." The grammar doesn't match up as is.

>>> Corrected (L 81).

4) Lines 93–94: “Data were available...”: Which data? All the data you used? I honestly was a little confused for a while if you were using some original data along with the data from previous studies. Maybe start the sentence “All sequence data used were...” to be clearer. Additionally, it would be very helpful to briefly describe how the data were obtained in the previous studies instead of just citing them.

>>> Clarified (L 94-98).

5) Lines 95–96: I got confused by what I think is incorrect grammar here. The commas seem to indicate a list, but are you only mentioning louse species and host species (2 items)? Which taxa does the “including samples across...” part refer to? Just lice, just hosts, or both? It needs to be clarified.

>>> Clarified (L 95-97)

6) Line 103: Why 28 as the phred score limit? I confess ignorance of this specific method when asking that question, but I feel like I wouldn't be the only one wondering. I'm not sure it's really necessary to explain, but it was definitely a question I had while reading.

>>> The standard procedure is to discard those bases with a phred quality score ≤ 20 (that equals to a 99 % base call accuracy). We opted for a more conservative approach and used 28 (that equals to a 99.8 %) as threshold. We have clarified this (L 104-105).

7) Lines 112–113: Okay, so this part really confused me, because I thought all your data came from previous studies. Which reads were newly generated and which were obtained from GenBank? I thought all were obtained from GenBank. The source(s) of all the data definitely needs to be clarified.

>>> Yes, we agree. We have clarified this aspect (L 111; 114).

8) Line 115: What does the “P” stand for in “P. emersoni”? Although I'm pretty sure you must mean *Physconelloides*, you've mentioned *Physconelloides* and *Pediculus* prior to this, so it would be good to clarify.

>>> Corrected (L 116).

9) Lines 124–125: Could be made clearer by deleting “both” and then inserting “between” in front of “individuals”

>>> Yes, done (L 126).

10) Line 130: Eliminate the use of “Finally” here because it appears between the “First” and “Second” methods described.

>>> Done (L 132).

11) Line 133: Spell out ILS because it's the first usage in the paper.

>>> Done (L 144).

12) Lines 135–136: Not that I think it would necessarily change results at all, but why trim the gene trees at this step instead of just construct the original gene trees with only the references? If you constructed the gene trees with only the reference taxa, would the PhyloNet results be the same?

>>> The logic behind our approach is that by trimming the gene trees after inference and not using only the same individuals for inference, the gene trees would be more accurately estimated than the other way around (i.e., because including more individuals in tree estimation will likely lead to a more accurate topology). Nevertheless, we agree that the results will probably be the same.

13) Lines 160–165: This is all redundant from the Methods section and feels like it was copy-and-pasted from a previous version of the Methods section (particularly because ILS is spelled out). It should be deleted from the Results section, in my opinion.

>>> Agree; done (L 172).

14) Line 175 (Figure 2, and Figs S1–S12): The supplementary figures were really good and helped me understand how the level of introgression was being measured for Figure 2. It might be worth putting one of the sppIDer graphs into the main paper for that purpose. In regards to the specific results here, some of them seem kind of unreal to me. Several of the *Columbicola* individuals had scores of >1, which means that over half of the reads from those individuals mapped to different species (if I'm understanding it correctly)?? Is that realistic? While the trend between the two genera clearly shows more introgression in *Columbicola* and makes sense, the specific values for introgression from the sppIDer results seem odd to me. Perhaps I do not understand the values well enough, but I think they could use some more explanation in the Discussion section. I can't be the only one that would see those results and wonder the same thing about how so much of a genome would map to different species.

>>> We agree, and we have included one example in the main text (and also one showing introgression across the set of loci). Regarding the introgression levels detected by sppIDer, we think they may be realistic, but we do not know for sure. We believe that much further research is needed to set the thresholds on what levels of introgression to expect (or would be realistic) in parasitic/symbiotic species. Having said this, sppIDer has been found to be very capable of retrieving introgression signatures for species missing in the dataset (Langdon et al., 2018). Thus, these introgression levels can certainly be aggregates of introgression events from different species that have not been including in the dataset. In other words, when we see a 5X mean coverage to a non-focal species this may be in fact introgression events to that species but also to others not included as a reference (e.g., extinct or those that typically inhabit other hosts and because of this we did not include it in our references). Indeed, the phylogenetic networks include several introgression events from ghost lineages and species that would support this scenario. On the other hand, we cannot rule out that some of these reads mapped to different species because of ILS (sppIDer does not control for ILS) or index-swapping. Nevertheless, we accounted for index-swapping in our statistical analyses, and both genera are expected to have relatively similar rates of ILS. In addition, PhyloNet, which does control for ILS, shows highly congruent results. Thus, we do not think these potential issues are biasing our conclusions on the role of dispersal shaping introgression rate. We have discussed this issue in the discussion (L 217-233).

15) Lines 183–184: This is the most important result that needs to be addressed before publishing, in my opinion. A chi-squared value of 3.8132 is not significant at the 0.05 level and would have a p-value of 0.05085. Either the p-value or chi-squared value listed here must be incorrect and needs to be fixed. If the chi-squared value is correct, then the discussion needs to be altered accordingly because the result is not as significant (but would still be good support for the hypothesis).

>>> We thank the reviewer for pointing this out. The confusion here is because the test is one-tailed. We used a one-tailed test because we were interested in testing our *a priori* prediction of whether *Columbicola* (the genus with higher dispersal capabilities) has a higher proportion of reticulations than *Physconelloides* (the genus with lower dispersal capabilities). The one-tailed test had higher statistical power to detect an effect in our relatively small species phylogenetic network dataset. Specifically, 0.54 vs. 0.42 statistical power to detect an effect size of 0.59 —estimated from the GLM analysis. We have now clarified the test we used and the hypothesis underlying each test in the manuscript (L 168-170; 188-192). Also, along with reporting the p-value for the two-tailed test (which is 0.05085 as the reviewer said), we have decided to report the statistics on effect size (i.e., R^2 for the GLM and confidence intervals for the difference in proportions), to clarify the magnitude of the effect we are reporting and thus to rely less on p-values (L 175-176; 188-192).

16) Lines 187–188: The third sentence of the caption is awkward. Perhaps eliminate the third sentence and change the second sentence to be “Orange branches depict reticulations, seven in *Columbicola* and four in *Physconelloides*”.

>>> Done (L 195).

17) Lines 196–197: “can provide further variation” -- variation in what?

>>> We meant genetic variation. Clarified (L 212).

“important eco-evolutionary consequences” -- Such as? This sentence could definitely use some fleshing out to be clearer about what you mean.

>>> Clarified (L 212-213).

Overall, I found the paper to be clearly written, aside from some minor sentence structure and grammar issues addressed above. And except for the redundant methods information at the beginning of the Results section, the manuscript does a very good job of being concise. The conclusions are strong without being oversold.

Methods-wise, I think enough detail has been provided to allow for reproduction, and I think the statistical analyses were sound (aside from the need to address the chi-squared results as reported).

Reviewers' comments:

Reviewer #1 (Remarks to the Author):

The revised manuscript is improved. The authors have addressed some, but not all of my comments.

1. I asked about the degree of divergence among the taxa being compared in each genus. The authors have responded by telling me the estimated timescale of the species divergence, which of course depends on their molecular clock assumptions. My question referred to the degree of genetic divergence. How does the degree of inter-taxa divergence compare to the level of intra-taxon diversity?

In fact, I am more inclined now to want to know more about the taxa being investigated. If the paper is to be of value to any readers other than those familiar with the pigeon feather lice literature, a little more explanation is required. When we get to Figure 4, at the end of the Results, we discover that some of the taxa are described with differing species names, but others are given the same species name. For example, why is *Physconelloides eurysema* 5 placed in the same species as *Physconelloides eurysema* 1-4, when it is depicted as being no more closely related to these than to *P. robbinsi* and *P. emersoni*?

2. I still find the nature of the evidence for introgression rather unclear, and the more I think about it, the more worried I become.

For example, from the description of the "level of introgression" (lines 132-134 and Figure 3 legend) I interpret that many of the *Columbicola* individuals are estimated to have more reads that map to other species, than map to the species to which the individual has been assigned; while on average, *Columbicola* individuals have only 50-60% of reads mapping to the genome of the species to which they are assigned. This purported level of introgression seems remarkably high? The one exceptional *Physconelloides* individual appears to have only ~20% of reads mapping to the genome of the species to which it has been assigned, and this certainly seems very odd.

Rather than simply relying on the somewhat black-box nature of the sppIDer output, I would think the authors could, from one individual, separately assemble the reads assigned to different species, to determine the nature of the regions being identified as introgressed. For example, typically how long are they?

And, as requested previously, they could show trees for particular genes demonstrating the highly inconsistent relationships among individuals, that must result from such massive levels of introgression. The authors have the data; why are they resistant to showing some phylogenies?

3. I suggested that the Figure (was Fig.3, now is Fig.4) showing reticulations was opaque. It remains so. The reticulation prior to the common ancestor of *P. eurysema* 1 and 2 is clear. But for several taxa (e.g., *P. robbinsi*), the only branch leading to them is in orange. How can this be a reticulation?

Supplementary Figures 13 and 14 show the reticulations. So the answer is that Figure 4 is not a good summary of the reticulations shown in the Supplement, and needs to be revised.

4. The authors have removed some of their claims for the significance of their results, but still claim that they "represent an important step towards understanding the factors driving hybridization, and have major implications for coevolutionary biology" (lines 33-34) and repeat this at line 234. I find this greatly overstated. Why not leave it to the reader to judge whether this is important or not?

Reviewer #1 (Remarks to the Author):

The revised manuscript is improved. The authors have addressed some, but not all of my comments.

>>> We appreciate the positive feedback about the revision and apologize for failing in addressing some of the reviewer comments. We have modified the manuscript and hope that these issues are now solved.

1. I asked about the degree of divergence among the taxa being compared in each genus. The authors have responded by telling me the estimated timescale of the species divergence, which of course depends on their molecular clock assumptions. My question referred to the degree of genetic divergence. How does the degree of inter-taxa divergence compare to the level of intra-taxon diversity?

>>> We have now included information on the distribution of intra -interspecific genetic distances (Table S3). Overall, *Columbicola* species have lower interspecific genetic distances. The same result was previously found by Sweet & Johnson (2018) that compared the degree of genetic divergence of two pairs of species of both genera inhabiting the same host species and sharing a cospeciation event. These differences are likely due to the higher gene flow rates consequence of their superior dispersal capabilities, differences in mutation rates, in timing of divergence, or some combination of these factors. Thus, even though PhyloNet does control by ILS, we cannot rule out that a fraction of the introgressed reads found by SpIDer may be caused by ILS. Accordingly, we have also expanded the discussion on the potential influence of ILS in SpIDer results (L 232-249).

In fact, I am more inclined now to want to know more about the taxa being investigated. If the paper is to be of value to any readers other than those familiar with the pigeon feather lice literature, a little more explanation is required. When we get to Figure 4, at the end of the Results, we discover that some of the taxa are described with differing species names, but others are given the same species name. For example, why is *Physconelloides eurysema* 5 placed in the same species as *Physconelloides eurysema* 1-4, when it is depicted as being no more closely related to these than to *P. robbinsi* and *P. emersoni*?

>>> We thank the reviewer for asking for this clarification —this point was unclear. This paper's taxonomic classification is based on Sweet & Johnson (2018) species delimitation analyses, who assigned numbers to putative species. The OTUs have not yet been officially described based on morphology. In particular, while most *Columbicola* OTUs (all but *C. passerinae* 1-2) matched known species, five *Physconelloides* OTUs (within *P. eurysema*) did not and are yet to be described as species. These undescribed *Physconelloides* OTUs are named here following Sweet & Johnson, 2018 (i.e., *P. eurysema* 1-5). We have clarified this issue (L 97-100).

2. I still find the nature of the evidence for introgression rather unclear, and the more I think about it, the more worried I become.

For example, from the description of the “level of introgression” (lines 132-134 and Figure 3 legend) I interpret that many of the *Columbicola* individuals are estimated to have more reads that map to other species, than map to the species to which the individual has been assigned; while on average, *Columbicola* individuals have only 50-60% of reads mapping to the genome of the species to which

they are assigned. This purported level of introgression seems remarkably high? The one exceptional *Physconelloides* individual appears to have only ~20% of reads mapping to the genome of the species to which it has been assigned, and this certainly seems very odd.

>>> We understand the reviewer's concerns. We also consider that this introgression level statistic from sppIDer is probably an overestimation of the real introgression level. In our opinion, it may overestimate introgression levels because this method: 1) does not control for ILS; 2) can detect introgression from species that are not included in the reference data, and thus the levels of introgression detected by sppIDer in certain species could be an aggregate of introgression events from more than one species; 3) misassigned reads (i.e., due to index-swapping). Even though we run simulations to demonstrate that this phenomenon was not biasing our comparative results, it may certainly be elevating the overall introgression levels in both groups. Overall, while we agree that the introgression levels detected by sppIDer may be unrealistic (and we acknowledge that in the Discussion), we do believe that the results of these analyses comparing the two genera are useful, because they are analyzed using the same techniques with the same assumptions. In addition, the results of the PhyloNet analysis, which does not share any of these problems, and is considered one of the best approaches to detect hybridization events, are highly congruent with those from sppIDer.

Rather than simply relying on the somewhat black-box nature of the sppIDer output, I would think the authors could, from one individual, separately assemble the reads assigned to different species, to determine the nature of the regions being identified as introgressed. For example, typically how long are they?

>>> For each library (i.e., sample), we have de-novo assembled all the reads mapping to other than the focal species. The assembled contigs are in size range of the loci used as a reference and, along with the coverage through genome plots, support they are not artificial mappings (e.g., high coverage mappings to short repetitive regions). We have included a supplementary table with the results of these assemblies (Table S3, L 142-145, 185-187).

And, as requested previously, they could show trees for particular genes demonstrating the highly inconsistent relationships among individuals, that must result from such massive levels of introgression. The authors have the data; why are they resistant to showing some phylogenies?

>>> We apologize for the misunderstanding. We had previously considered that PhyloNet analyses, which summarized individual gene tree histories and provided information about hybridization, was sufficient. To some extent, focusing on particular introgressed genes was beyond the scope of this paper. However, as requested, we have now plotted and included some example gene trees (Figs. S13-S14) that illustrate gene tree signatures of introgression in the supplementary material and deposited all the individual gene trees in Figshare. These trees show evidence that some introgressed individuals share very similar sequences in some genetic loci with other species.

3. I suggested that the Figure (was Fig.3, now is Fig.4) showing reticulations was opaque. It remains so. The reticulation prior to the common ancestor of *P. eurysema* 1 and 2 is clear. But for several taxa (e.g., *P. robbinsi*), the only branch leading to them is in orange. How can this be a reticulation?

Supplementary Figures 13 and 14 show the reticulations. So the answer is that Figure 4 is not a good summary of the reticulations shown in the Supplement, and needs to be revised.

>>> We agree. We have replaced figure 4 with a modified version of Figure S13 and S14 (L 206).

4. The authors have removed some of their claims for the significance of their results, but still claim that they “represent an important step towards understanding the factors driving hybridization, and have major implications for coevolutionary biology” (lines 33-34) and repeat this at line 234. I find this greatly overstated. Why not leave it to the reader to judge whether this is important or not?

>>> We agree and have removed these claims. We have also removed another subjective statement from the discussion (“will certainly have a strong impact in coevolutionary biology theory”).